# Functional Analysis of Long Non-Coding RNAs Reveal Their Novel Roles in Biocontrol of Bacteria-Induced Tomato Resistance to *Meloidogyne incognita*

**DOI:** 10.3390/ijms21030911

**Published:** 2020-01-30

**Authors:** Fan Yang, Dan Zhao, Haiyan Fan, Xiaofeng Zhu, Yuanyuan Wang, Xiaoyu Liu, Yuxi Duan, Yuanhu Xuan, Lijie Chen

**Affiliations:** 1College of Plant Protection, Shenyang Agricultural University, Dongling Road 120, Shenyang 110866, China; yangjingdong2333@163.com (F.Y.); fanhaiyan6860@gmail.com (H.F.); syxf2000@syau.edu.cn (X.Z.); duanyx@syau.edu.cn (Y.D.); xuanyuanhu115@syau.edu.cn (Y.X.); 2College of Plant Protection, Jilin Agricultural University, Xincheng Road 2888, Jilin 130118, China; zhaodan1201@jlau.edu.cn; 3College of Biotechnology, Shenyang Agricultural University, Dongling Road 120, Shenyang 110866, China; wangyuanyuan2018@gmail.com; 4College of Science, Shenyang Agricultural University, Dongling Road 120, Shenyang 110866, China; liuxiaoyu7805@163.com

**Keywords:** long non-coding RNA, biocontrol bacteria, *Meloidogyne incognita*, tomato, induced resistance

## Abstract

Root-knot nematodes (RKNs) severely affect plants growth and productivity, and several commercial biocontrol bacteria can improve plants resistance to RKNs. *Pseudomonas putida* Sneb821 isolate was found to induce tomatoes resistance against *Meloidogyne incognita*. However, the molecular functions behind induced resistance remains unclear. Long non-coding RNA (lncRNA) is considered to be a new component that regulates the molecular functions of plant immunity. We found lncRNA was involved in Sneb821-induced tomato resistance to *M. incognita*. Compared with tomato inoculated with *M. incognita*, high-throughput sequencing showed that 43 lncRNAs were upregulated, while 35 lncRNAs were downregulated in tomatoes previously inoculated with Sneb821. A regulation network of lncRNAs was constructed, and the results indicated that 12 lncRNAs were found to act as sponges of their corresponding miRNAs. By using qRT-PCR and the overexpression vector pBI121, we found the expression of lncRNA44664 correlated with miR396/GRFs (growth-regulating factors) and lncRNA48734 was correlated with miR156/SPL (squamosal promoter-binding protein-like) transcription factors. These observations provided a novel molecular model in biocontrol bacteria-induced tomato resistance to *M. incognita*.

## 1. Introduction

Non-coding RNAs (ncRNAs) play key roles in biological processes by regulating gene expression at various stages, including at transcriptional, translational and epigenetic levels [1,2,3]. Only a few ncRNAs were confirmed to exist in plant cells, including small interfering RNAs (siRNAs), microRNAs (miRNAs), trans-acting siRNAs (tasiRNAs) and long non-coding RNAs (lncRNAs) [4,5,6,7]. lncRNAs are typically longer than 200 nucleotides and primarily transcribed by RNA polymerase, which do not encode the open reading frame (ORF). lncRNAs play important functional roles in regulation by forming networks of ribonucleoprotein complexes with chromatin regulators and targeting their action to appropriate genomic regions [8]. Study of lncRNAs is becoming a novel research area in plant resistance. For example, lncRNA33732 has the potential to affect *RBOH* gene expression to improve plant resistance to disease [9]. Furthermore, lncRNA23468 has the potential to regulate the expression of its target gene *NBS-LRR* [10]. Therefore, lncRNAs may play a role in regulating plant disease resistance.

Root-knot nematodes (RKNs) are widespread and have an extremely broad host range that comprise many plant species [11]. Among all RKN species, *M. incognita* is the most widely spread and highly studied [12]. Tomato (*Solanum lycopersicum* L.) is among the most economically important vegetable crops and it can be infected by RKNs [13]. With the enhancement of molecular research, studies of non-coding RNAs and RKNs have been investigated. The miR319/TCP4 module is essential in tomato galls to modulate the Jasmonate acid (JA) biosynthesis induced by RKNs invasion [14]. It has been previously shown that miR159 plays a role in gall development by inhibiting *MYB33* translation [15]. Studies of miR390a and *tas3* loss-of-function mutants reported the production of fewer galls, suggesting that the miR390/TAS3/ARF3 regulatory module is required for correct gall formation [16]. The regulatory gene module composed by miR172 and the two transcription factors TOE1 (target of early activation tagged 1) and FT (flowering locus T) have been demonstrated in root galls during the formation of giant cells in *Arabidopsis* [17]. lncRNAs act as a miRNA sponge that contain miRNA-binding sites, which have been indicated to regulate miRNAs and their targets [18]. miRNAs have been primarily explored, but lncRNAs in plants inoculated with RKNs have not previously been reported. 

Plant growth-promoting rhizobacteria (PGPR) are defined as biocontrol bacteria that protect plants against RKNs. Our previous studies have shown that *Pseudomonas putida* Sneb821 induce tomatoes resistance to *M. incognita* [13]. *Pseudomonas* MLR6 inoculation provided a positive effect on cell membrane stability by reducing the electrolyte leakage and priming the reactive oxygen species (ROS) accumulation [19]. Another study showed that *Pseudomonas aeruginosa* was able to switch from a planktonic to a sessile lifestyle, as the two-component system (TCS) GacS/GacA activated the production of two small ncRNAs, RsmY and RsmZ [20]. ncRNAs are involved in the process of biocontrol bacteria-induced plant resistance to diseases, but the molecular functions of ncRNAs remains unclear, specifically at the lncRNA level. 

In the present study, we explored that Sneb821 induced tomatoes resistance to *M. incognita* by regulating plants ROS responses. RNA sequencing was performed and analysis of lncRNAs was carried out between tomatoes inoculated with *M. incognita* or biocontrol bacteria and *M. incognita*. A competing endogenous RNA (ceRNA) network was also constructed to demonstrate the interaction of lncRNAs, miRNAs and mRNAs. qRT-PCR and two overexpression vectors were used to explore the functions of those lncRNAs. Our results suggested that lncRNAs act as a regulatory factor in biocontrol bacteria to induce tomatoes resistance against *M. incognita* infection.

## 2. Results

### 2.1. Inducement Mechanisms of Sneb821

The strain of *P. putida* Sneb821 was isolated and identified. Sneb821 displayed plant growth-promoting traits as well as nematocidal activity and was found to be efficient in controlling *M. incognita* in tomatoes [13]. This study involved investigating the mechanisms of these bacteria as biological control agents of RKNs. A split root experiment was carried out to prove induced resistance of Sneb821 (Figure 1A). Sneb821 treatment caused slow development of nematodes in the roots, and only 14.5% of second-stage juveniles (J2) developed into third-stage juveniles (J3), which was significantly different from the control group (Figure 1B). The results showed that the disease resistance mechanism of Sneb821 is induced resistance. The inducement mechanism of Sneb821 was explored. In previous studies, infection of Arabidopsis with *Pseudomonas* induced a burst of ROS. *Pseudomonas syringae* DC3000, infected with Arabidopsis leaves, induced the target gene required to participate in plant defense responses to pathogens [21]. Two experiments were set up to explore the inducement mechanism of Sneb821 in two treated groups including inoculation of *M. incognita* treated plants (RKNs treatment, RKNT) group and inoculation of *M. incognita* treated plants after Sneb821 irrigation 5 days (Sneb821 treatment, Sneb821T) group. NB liquid medium was used as the control. The contents of H_2_O_2_ and O_2_^−^, and expression level of *RBOH1* genes at 1 dpi, 3 dpi, 6 dpi, 12 dpi in RKNT and Sneb821T was determined. The H_2_O_2_ content of Sneb821T was significantly higher than RKNT at 3 dpi and 6 dpi (Figure 1C). The O_2_^−^ production of Sneb821T was significantly higher than RKNT at 3 dpi, 6 dpi and 12 dpi (Figure 1D). Furthermore, the relative expression of *RBOH1* gene was significantly higher than RKNT at 3 dpi and 6 dpi (Figure 1E). These results revealed that biocontrol bacteria induced upregulation of ROS-related responses in plants.

### 2.2. Sequencing and Characteristics of Transcripts

A comprehensive transcriptomic profiling was performed to dissect the molecular mechanism associated with plant induced resistance to *M. incognita*. Over 218.47 million raw reads were produced from six cDNA libraries of tomato samples after removing adaptor sequences and low-quality reads. The mapped reads and mapped ratio of RKNT1, RKNT2, RKNT3 were (108742693 (88.36%), 105048103 (89.01%), 114727340 (88.84%), respectively). Sneb821T1, Sneb821T2, Sneb821T3 were mapped (104203875 (89.86%), 114918861 (90.71%), 99030869 (90.54%) respectively) and reads to the reference genome of tomato (Table 1), the lncRNAs were mapped on to the tomato genome SL2.50. In total, 3371 lncRNAs were predicted with CNCI, CPC, Pfam and CAPT (Figure 2A). The classification of the predicted novel lncRNAs showed that 51 were intronic lncRNAs, 601 were antisense lncRNAs, 731 were sense lncRNAs and 1988 were long intervening/intergenic non-coding RNAs (lincRNA) (Figure 2B). The structural feature and expression analysis of lncRNAs have been shown in Appendix A.

To identify additional RKN-related lncRNAs, the expression levels were compared and total lncRNAs were differentially expressed between RKNT and Sneb821T samples. The heat map showed the expression pattern of lncRNAs between the two groups (Figure 2C). Compared with RKNT group, 43 lncRNAs were upregulated, while 35 lncRNAs were downregulated in tomato Sneb821T (Figure 2D).

In these differentially expressed lncRNAs, we found 78 lncRNAs (Appendix A). These lncRNAs may play a key role in bacteria-tomato-RKNs interactions. In order to confirm the reliability of the predicted lncRNAs in tomatoes, 12 lncRNAs were selected for RT-PCR and successfully amplified (Appendix A, the primer sequence is shown in Appendix A), suggesting that a high percentage of predicted lncRNAs are reliable in terms of expression. Therefore, our data can be used for future research, and 12 new lncRNAs were involved in the process by which Sneb821 induces tomato resistance to *M. incognita*.

### 2.3. Gene Ontology (GO) and Kyoto Encyclopedia of Genes and Genomes (KEGG) Analysis

To reveal potential functions of lncRNA candidates, their co-localization genes and DEGs were identified within the neighboring 100 kb region and GO terms and KEGG pathway enrichment of these genes were analyzed. The GO terms are shown in Figure 3A, with a total of 15572 neighboring genes and 408 DEGs were assigned to at least one GO term. The most highly represented categories within the GO molecular function were ‘catalytic activity’ (201 genes) and ‘binding’ (188 genes). For the GO cellular components, genes involved in ‘cell’ (179 genes) and ‘cell part’ (180 genes) were the most highly represented. For the biological processes, the two highest represented categories were ‘metabolic process’ (274 genes) and ‘cellular process’ (248 genes). The most highly represented categories within the GO molecular function, cellular components and biological processes were almost identical to biocontrol bacteria which induced tomato resistance to *M. incognita* (Appendix A). The process of Sneb821 inducing tomato resistance to *M. incognita* affected the expression of genes involved in those pathways.

In the KEGG database, these pathways were also distributed in five major categories, including metabolism, organismal systems, environmental information processing, cellular processes and genetic information processing (Figure 3B). Common enrichments were observed in carbon metabolism (11 genes), biosynthesis of amino acids (10 genes), plant-pathogen interaction (5 genes), plant hormone signaling system (9 genes), endocytosis (6 genes), ribosome (16 genes) and RNA transport (10 genes). lncRNA21563, lncRNA44664 and lncRNA48734 were related to plant hormone signal transduction (ko04075), cell proliferation (GO:0008283) and oxidative phosphorylation (ko00190). The target genes of selected 12 lncRNAs for GO terms and KEGG pathways are shown in Table 2. The results showed that those pathways may be associated with induced disease resistance in tomatoes (Appendix A).

### 2.4. Construction of the CeRNA Network and Predicted Interaction between lncRNAs, miRNAs, and mRNAs

To explore the functions of the lncRNAs target genes, the ceRNA network of these 12 lncRNAs was mapped, as lncRNA can affect mRNA through miRNAs and play a role in regulating expression of genes (Figure 4A). This study annotated all target genes and firstly found that lncRNA44664 and lncRNA48734 has the potential to regulate miR396a and miR156d, respectively, to affect plants responses to *M. incognita*. The gene annotation result showed that Solyc10g083510.1.1 was annotated as a growth-regulating factor, and Solyc05g012040.2.1 was annotated as a squamosa promoter binding protein (Table 3). The ceRNA regulatory networks may help explain the ability of lncRNAs in Sneb821 to induce tomato resistance to *M. incognita*.

Interplay between miRNAs and lncRNAs is an important functional pattern seen for lncRNAs [22]. lncRNAs could be targeted by miRNAs [23] and could also function as endogenous target mimics (eTMs) of miRNAs [18]. To examine whether lncRNAs are true targets for miRNAs, 12 lncRNAs were assessed using psRNATarget and psRobot. The interactions and connection sites between miRNAs and lncRNAs were identified (Figure 4B). An association between miRNAs and lncRNAs was identified. The interaction between mRNAs and miRNAs was also identified (Figure 4C). These results strongly suggest that the ceRNA network is reliable and the regulations of 10 miRNAs may have occurred from the alteration level of 12 lncRNAs.

### 2.5. Verification of Differentially Expressed lncRNAs

To validate the expression of lncRNAs in tomatoes after *M. incognita* infection, two lncRNAs (lncRNA44664 and lncRNA48734) and their target genes were selected for verification. qRT-PCR in root samples were carried out to verify the sequencing results. The expression of lncRNA44664 and its target gene *Solyc10g083510.1* of Sneb821T was significantly lower than RKNT at 6 dpi (Figure 5A,B), with the expression of miR396a of Sneb821T being significantly higher than RKNT at 6 dpi (Figure 5B). The expression of lncRNA48734 was significantly lower than RKNT at 3 dpi and 6 dpi (Figure 5C), with the expression of miR156d of Sneb821T being significantly higher than RKNT and its target gene *Solyc05g012040.2* of Sneb821T was significantly lower than RKNT at 6 dpi (Figure 5D). These results showed that the identified lncRNA44664 and lncRNA48734 regulated miR396 and miR156, respectively, in tomatoes. Furthermore, it suggests that these lncRNAs may play important roles in Sneb821-induced plant response to *M. incognita* infection.

### 2.6. Function Verification of lncRNAs

To verify the functions of lncRNAs, the relationship between the lncRNAs, miRNAs and its target genes, using infiltration to upregulate lncRNA44664 and lncRNA48734 expressions in MoneyMaker tomatoes, was investigated. The overexpression of lncRNA 44664 and lncRNA48734, using plasmids, was carried out on the basis of the pBI121 vector (Figure 6A). The introduction of A. tumefaciens harboring pBI121-lncRNA44664 and pBI121-lncRNA48734 into tomato leaf cells resulted in significant upregulation of lncRNA44664 and lncRNA48734 expression at 3 dpi. The expression levels of lncRNA44664 and lncRNA48734 were greater than the tomato leaves that overexpressed the empty vector (EV) (Figure 6B,C). The expression of miR396a was suppressed to approximately 50% in the tomato plants that overexpressed lncRNA44664 (OE-lnc44664), and the target genes of *Solyc10g083510.1* were significant increased (Figure 6B). miR396 was found to be associated with GRFs (growth-regulating factors) [24]. The expression of miR156d was suppressed to approximately 50% in the tomato plants that overexpressed lncRNA48734 (OE-lnc48734), and the target genes of *Solyc05g012040.2* were significantly increased (Figure 6C). miR156 was found to be associated with SPL (squamosal promoter-binding protein-like) transcription factors [25]. These results strongly suggested that there is an interdependency between lncRNA44664-miR396-GRFs- and lncRNA48734-miR156-SPL-mediated processes of biocontrol bacteria inducing plants response to RKNs infection.

The possible function of lncRNA48734 was also investigated. The leaves of OE-lnc48734 and EV tomato plants were used to detect the accumulation of H_2_O_2_ by 3,3′-diaminobenzidine (DAB) staining. After 3 days of *Agrobacterium* infiltration, dark brown-colored polymeric oxidation products (from H_2_O_2_) accumulated in the surrounding area of the leaves of EV plants, and smaller dark brown color was observed in the surrounding area of the leaves of OE-lnc48734 plants (Figure 6D). miR398 was found to be regulated by SPL transcription factors [26]. *CSD* (Cu/Zn-superoxide dismutase) gene was a target gene of miR398, as it could regulate the synthesis of superoxide dismutase (SOD) [27], and SOD was the main antioxidant enzyme that scavenges superoxide, it can catalyze O_2_^−^ to H_2_O_2_ [28]. The expressions of *CSD1* and *CSD2*, the target gene of miR398, were also measured. The results showed that the expression of *CSD1* was significantly decreased but *CSD2* was not significantly altered (Figure 6E). The mode of lncRNA48734 and H_2_O_2_ interaction was shown in Figure 6F. These results suggested that lncRNA48734 might have a negative role in H_2_O_2_ production as it played a novel role in the process of biocontrol bacteria inducing plant response to *M. incognita* infection.

## 3. Discussion

### 3.1. ROS Participate in Sneb821-induced Tomato Resistance to M. incognita

When plants are inoculated with biocontrol bacteria, its defense system is activated and causes the induction of a series of defensive responses, such as a hypersensitive response (HR) and ROS burst. *Pseudomonas* has the potential to induce ROS of *Arabidopsis* to burst [21]. *Pseudomonas* MLR6 inoculation could cause ROS accumulation in plants [19]. A split root experiment was carried out to prove the induced resistance of Sneb821, and this study has shown that the *RBOH1* gene was upregulated and contents of H_2_O_2_ and O_2_^−^ were increased in bacteria-induced tomato resistance to *M. incognita* (Figure 1), compared with controls. These results strongly suggested that reactive oxygen-related pathways may play an important role in Sneb821-induced tomato resistance to *M. incognita*.

### 3.2. Certain lncRNAs Participate in Sneb821-induced Tomato Resistance to M. incognita

Our research was the first of its kind to discover that lncRNAs play a role in biocontrol bacteria-induced tomato resistance to *M. incognita*. A few studies have reported the role of lncRNA in other plant pathogens. Research has shown that certain lncRNAs may affect *RBOH* gene expression to improve plant resistance to disease [9], and lncRNAs may be involved in plant resistance. Although studies of lncRNAs are in the early stages, an increasing number of lncRNAs have been characterized and found to play key roles in various resistance processes [29]. Expression analysis revealed that lncRNAs play a crucial role in the resistance mechanism of wheat, indicating mechanisms that regulate defense pathways to stripe rust [30]. In tomatoes, 688 differentially expressed lncRNAs were identified between *Phytophthora infestans*-resistant and -susceptible tomato lines, to enhance tomato resistance to *P. infestans* [31]. The rice lncRNAs in response to the infection of *Xanthomonas oryzae* pv. was identified, and the regulation of lncRNA-mediated JA pathway on rice resistance was described [32]. Some lncRNAs have recently been identified in plants but limited studies have investigated whether biocontrol bacteria induce plant resistance to diseases. In our study, an initial genome-wide identification of lncRNAs in tomatoes inoculated with biocontrol bacteria or *M. incognita* was carried out by high-throughput sequencing technology. In total, 3771 lncRNAs were found and 43 lncRNAs were upregulated, while 35 lncRNAs were downregulated in bacteria-induced tomato resistance to RKNs (Figure 2). Our study confirmed the expression of 12 lncRNAs in tomatoes by RT-PCR assay (Appendix A), demonstrating the existence of a high proportion of lncRNAs that was predicted. The results suggested that these novel lncRNAs participated in the process of Sneb821-induced tomato resistance to *M. incognita*.

In a previous lncRNA study, biological processes such as salicylic acid metabolic processes, fatty acid biosynthetic processes and nitrate transport were the most highly represented in GO terms and 20 pathways were enriched in KEGG pathway enrichments [33]. Moreover, our results showed that GO terms and KEGG pathway enrichments analysis demonstrated the target genes were associated with a variety of metabolic terms and pathways, such as ‘catalytic activity’ (201 genes), ‘cell’ (179 genes) and ‘cell part’ (180 genes) terms. Plant-pathogen interaction (5 genes) and plant hormone signaling system (9 genes) were enriched in the KEGG pathway. Furthermore, the peroxisome pathway (2 genes) was also enriched (Figure 3). GO terms and KEGG pathways for the target genes of selected 12 lncRNAs are shown in Table 2. lncRNA44664 and lncRNA48734 were related to cell proliferation and oxidative phosphorylation. These bioinformatic results may help us to explain the induced mechanism of Sneb821 and the mechanisms behind how lncRNAs regulate gene expression through the cell-related and pathogen-related pathways.

### 3.3. Inducement Resistance Function in lncRNAs through the ceRNA Network

Twelve lncRNAs and related miRNAs and mRNAs were used to build a ceRNA network. The network was first established in the induced resistance study. eTMs with miRNAs is a regulatory mechanism of lncRNA. In *Arabidopsis*, 36 lncRNAs were identified as eTMs for 11 conserved miRNAs, and eTMs of miR160 and miR166 are functional in the regulation of plant development [18]. In rice, several lncRNAs were also identified as competing endogenous RNAs, which bound miR160 and miR164 in a type of target mimicry [34]. In a previous study, tomato lncRNAs, lnc0195 and lnc1077, acted as competing eTMs for tomato miR166 and miR399, respectively [34]. In our study, the ceRNA network of these lncRNAs, miRNAs and mRNAs was mapped to elaborate their role in bacteria-induced tomato resistance to *M. incognita* (Figure 4 and Table 3). These results indicated that the mRNA-miRNA-lncRNA pair might be an important regulatory pattern in bacteria-induced tomato resistance to *M. incognita*.

### 3.4. lncRNA48734 and lncRNA44664 in Sneb821-induced Tomato Resistance to M. incognita

In our study, lncRNA48734 was predicted to be a ‘decoy’ for tomato miR156/SPL, and lncRNA44664 correlated with miR396/GRFs. In plants, overexpression of lncRNA33732 increased expression of *RBOH* gene, and silencing of lncRNA33732 resulted in decreased expression of *RBOH* gene, suggesting that lncRNA33732 may affect *RBOH* gene expression [9]. Studies have also found that when lncRNA23468 is overexpressed in tomatoes, the expression of its target gene *NBS-LRR* was significantly increased, and disease resistance was enhanced. In our study, lncRNA48734 was predicted to be associated with miR156/SPL (Figure 5), and SPL was shown to be a transcription factor of miR398. miR398 downregulates *CSD* gene, which was associated with H_2_O_2_ production [28]. Our study also predicted that lncRNA44664 was associated with miR396/GRFs (Figure 5). Previous studies demonstrated that miR396 was significantly downregulated in response to cyst nematodes [35], which is consistent with the present results. Furthermore, miR396 overexpression reduced the syncytium size and arrested cyst nematode development by repressing GRFs in *Arabidopsis* roots [36]. By using an overexpression vector, lncRNA44664-miR396-GRFs- and lncRNA48734-miR156-SPL-mediated patterns were identified in biocontrol bacteria-induced plants response to *M. incognita* infection. Moreover, lncRNA48734 has been associated with reactive oxygen regulation and lncRNA44664 may be associated with GRFs (Figure 6). Therefore, lncRNA48734 and lncRNA44664 may reveal the molecular mechanisms behind biocontrol bacteria regulating plant resistance to *M. incognita* (Figure 7). However, future research is needed to investigate the link between bacteria, lncRNAs, GRFs and *M. incognita*. 

Currently, research in plant-pathogen interaction is advanced, and this study explored the interaction between plant-biocontrol bacteria-pathogen. This study showed that lncRNAs play a role in the plant-biocontrol bacteria-pathogen interaction. Furthermore, we confirmed the novel lncRNAs functions in bacteria-induced plant resistance to *M. incognita*. Our findings not only enhance the lncRNAs database for tomatoes, but also lay a foundation for further investigations of potential lncRNA-mediated regulation of bacteria-induced plant resistance to *M. incognita*.

## 4. Materials and Methods 

### 4.1. Plant Materials, Biocontrol Bacteria and Nematode Inoculum

Tomato seeds (*Solanum lycopersicum* cultivar MoneyMaker, susceptible to RKNs), with all seeds treated with bacterial fermentation broth (10^8^ CFU/mL) at the rate of 20 μL/g for 5 min at 25 °C, with NB (nutrient broth) liquid medium used as the control. Eggs of *M. incognita* were extracted from the roots of tomatoes and surface-sterilized with 10% sodium hypochlorite for 3 min. The eggs were hatched in sterile water at 26 °C, and J2s were collected every 24 h. The nematode suspensions were diluted to 200 ± 10 J2/mL with distilled water and stored at 15 °C for subsequent experiments. Three days after biocontrol bacteria inoculum, 2000 freshly hatched RKNs *M. incognita* were inoculated into each pot. Five days after inoculation with *M. incognita*, tomato RNA was extracted.

### 4.2. RNA Extraction and Library Construction

Trizol (Invitrogen, USA) was used for extraction of total RNA from tomato inoculated RKNs and control tomato, according to manufacturer’s instructions. The RNA concentration and purity were checked by OD A260/A280 (>1.8) and A260/A230 (>1.6), and the yield and quality were assessed using an Agilent 2100 Bioanalyzer (Agilent Technologies, California, USA) and RNA 6000 Nano LabChip Kit (Agilent Technologies, California, USA). Clean data were obtained after removing reads containing adapter, reads containing ploy-N and low-quality reads from raw data. Meanwhile, Q20, Q30, GC-content and sequence duplication level of the clean data were calculated. All downstream analyses were based on clean data with high quality. Original sequencing data have been uploaded to the National Centre for Biotechnology Information (NCBI) as BioProject PRJNA592450.

### 4.3. lncRNA Analysis

The assembled transcripts were annotated using the gffcompare program. The unknown transcripts were used to screen for putative lncRNAs. Four computational approaches including CNCI [37], CPC [38], Pfam [39] and CAPT [40] were combined to sort non-protein coding RNA candidates from putative protein-coding RNAs in the unknown transcripts. Putative protein-coding RNAs were filtered out using a minimum length and exon number threshold. Transcripts with lengths more than 200 nt and with more than two exons were selected as lncRNAs candidates and further screened, using CNCI, CPC, Pfam and CAPT softwares, as they have the power to distinguish the protein-coding genes from the non-coding genes.

### 4.4. RT-PCR Validation

Total RNA of tomato samples was extracted with the AxyPre RNA extraction kit (Axygen, California, USA) following the manufacturer’s protocol. Total RNA recovered was immediately used to generate first strand cDNA using the PrimeScript cDNA synthesis kit (TAKARA, Dalian, China). miRNA First Strand cDNA Synthesis Tailing Reaction (Sangon Biotech, Shanghai, China) was used to generate first strand cDNA, according to the manufacturer’s instructions. Synthesized cDNA was stored at −20 °C. The specific primers were designed with DNAMAN software (Lynnon Biosoft, CA, USA) and synthesized by Sangon Biotech Co., Ltd. (Shanghai, China). The primer sequences with successful amplification were presented in Appendix A. PCR was conducted on a T100 thermo cycler (BIO-RAD, California, USA) using Premix (TAKARA, Dalian, China), according to the manufacturer’s protocol (pre-denaturation step at 94 °C for 5 min; 30 amplification cycles of denaturation at 94 °C for 50 s, annealing at 55 °C for 30 s, and elongation at 72 °C for 1 min; followed by a final elongation step at 72 °C for 10 min). The PCR products were examined on a 1.5% agarose gel with Genecolor (Gene-Bio, Beijing, China) staining.

### 4.5. Quantitative Real-Time PCR Validation

After RNA isolation from the root of tomatoes, qRT-PCR was performed using SYBR Premix Ex Taq (Takara, Dalian, China) and MicroRNAs qPCR Kit (Sangon Biotech, China), according to the manufacturer’s instructions. Subsequently, qRT-PCR was performed in a 20 μL reaction volume, including 10 μL SYBR Green Master Mix, 0.8 μL PCR Forward Primer (10 μM), 0.8 μL PCR Reverse Primer (10 μM), 0.4 μL ROX, 2 μL cDNA and 6 μL nuclease-free water. The protocol was initiated at 95 °C for 5 min, followed by 95 °C (5 s) and 60 °C (34 s) for a total of 40 cycles. β-actin was used as a reference gene. Reactions were performed in three independent wells. The 2^−ΔΔCt^ method [41] was used to calculate the relative RNA expression levels. Student’s *t*-tests were performed, and results were considered significant when *p* < 0.05. The values were expressed as means ± standard deviation (SD). The primers sequences are shown in Appendix A.

### 4.6. Prediction of lncRNA Targets and miRNA eTMs from mRNA

Tomato lncRNA and mRNA were predicted as miRNA targets using the psRNATarget [42] and psRobot [43]. The miRNA eTMs from tomato lncRNAs were predicted based on a previous study [18]. Phytozome (https://phytozome.jgi.doe.gov/pz/portal.html) was used to annotate tomato target genes.

### 4.7. ceRNA Network Analysis

TargetFinder software [44] was used to predict target miRNAs of lncRNAs and target mRNAs of miRNAs. The default of prediction score cutoff value was set at seven. The input files are mRNA, miRNA and lncRNA FASTA sequences files. Cytoscape (https://cytoscape.org/) was used to draw a ceRNA network.

### 4.8. GO and KEGG Pathway Analysis

GO analysis (http://www.geneontology.org) was performed to construct gene annotations. KEGG pathway analysis was performed to understand the function and interactions among differentially expressed genes. The enrichment factor was the value ratio between the sequenced gene and all annotated genes enriched in the pathway.

### 4.9. Transient Overexpression and Agrobacteria Infiltration

According to the tomato genome and lncRNA prediction results, primers were designed and used to clone lncRNA44664 and lncRNA48734 from tomato plants (Appendix A). The PCR fragments of lncRNA44664 and lncRNA48734 were subcloned into the binary vector, pBI121, replacing the *GUS* gene. In plasmids, lncRNA44664 and lncRNA48734 were controlled by the Cauliflower mosaic virus (CaMV) 35S promoter. All of the plasmids were transformed into *Agrobacteria tumefaciens* strain GV3101 by the freeze-thaw method [45].The plant was infiltrated with the *Agrobacteria* infiltration using a syringe [46]. *A. tumefaciens* containing pBI121-lncRNA44664 and pBI121-lncRNA48734 plasmids was introduced into the leaves of MoneyMaker tomatoes by infiltration. *A. tumefaciens* with an empty vector was used as a control. The transient overexpression plants had five biological repetitions at least. The leaves were harvested for the observation at 3 dpi (days post infection).

### 4.10. ROS Level Determination and DAB Staining

The contents of H_2_O_2_ determination was calculated using a Hydrogen Peroxide Contents Detection Kit (Solarbio, Beijing, China) and O_2_^−^ production determination was calculated using a Micro Superoxide Anion Assay Kit (Solarbio, Beijing, China) [47]. 3,30-Diaminobenzidine (DAB) was used to measure the levels of H_2_O_2_, as described in a previous study [48]. All experiments have been repeated independently three times with similar results.

## 5. Conclusions

Our study has illustrated that lncRNAs play important roles in the plant-biocontrol bacteria-pathogen interaction. Genome-wide identification of lncRNAs in tomatoes and analysis of their functions suggested that the selected 12 lncRNAs play a regulated role in the process of bacteria-induced plant resistance to *M. incognita*. Moreover, we constructed the ceRNA network of 12 lncRNAs and investigated the relationship of lncRNAs, miRNAs and parent genes. Finally, further functional characterization of lncRNA48734 and lncRNA44664 elucidated the lncRNA44664-miR396-GRFs- and lncRNA48734-miR156-SPL-mediated functions for these lncRNAs. The results showed that lncRNA48734 and lncRNA44664 may be regulated in the process of bacteria-induced plant resistance to *M. incognita*. Our approach may therefore be valuable for detecting bio-control mechanism of plant defense, discovering new biomarkers for the resistance of plants to *M. incognita* and exploring the pathogenic functions of this plant disease.

## Figures and Tables

**Figure 1 ijms-21-00911-f001:**
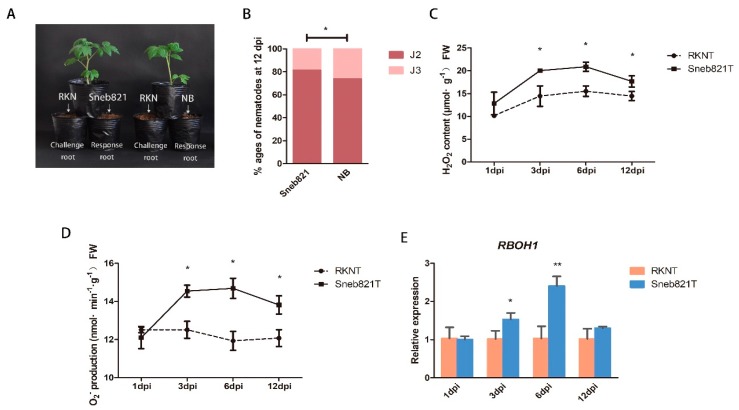
Inducement mechanisms of *P. putida* Sneb821. (**A**) The split root experiment of Sneb821. (**B**) Percentage of nematodes in different developmental ages at 12 days post inoculation (dpi). (**C**) The H_2_O_2_ content of Sneb821T and RKNT. (**D**) The O_2_^−^ production of Sneb821T and RKNT. (**E**) The relative expression of *RBOH1* gene of Sneb821T and RKNT. All data are the means ± SD of three independent experiments. * indicate a significant difference at the *p* = 0.05 level, ** indicate a significant difference at the *p* = 0.01 level. RKN means root knot nematode treatment, NB means nutrient broth liquid medium treatment, RKNT means RKNs treatment, Sneb821T means RKNs and Sneb821 treatment.

**Figure 2 ijms-21-00911-f002:**
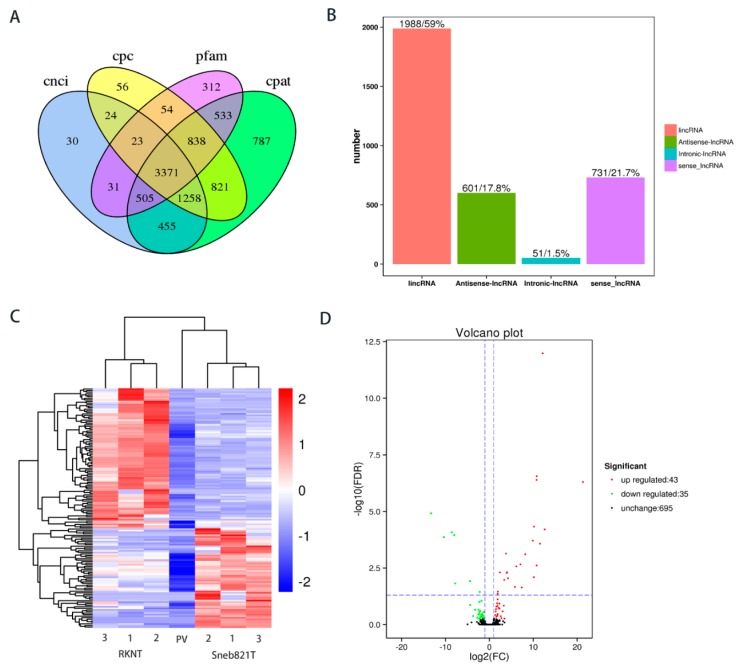
Prediction and difference analysis of novel lncRNA of tomato. (**A**) Venn analysis of the predicted novel lncRNAs from CNCI, CPC, Pfam and CAPT softwares. (**B**) Classification of the predicted novel lncRNAs. (**C**) Heat map in tomato RKNT and Sneb821T. 1, 2, 3 = three independent experiments. PV = *p*-Value. (**D**) Volcano plot in tomato RKNT and Sneb821T. FC = fold change; FDR = false discovery rate.

**Figure 3 ijms-21-00911-f003:**
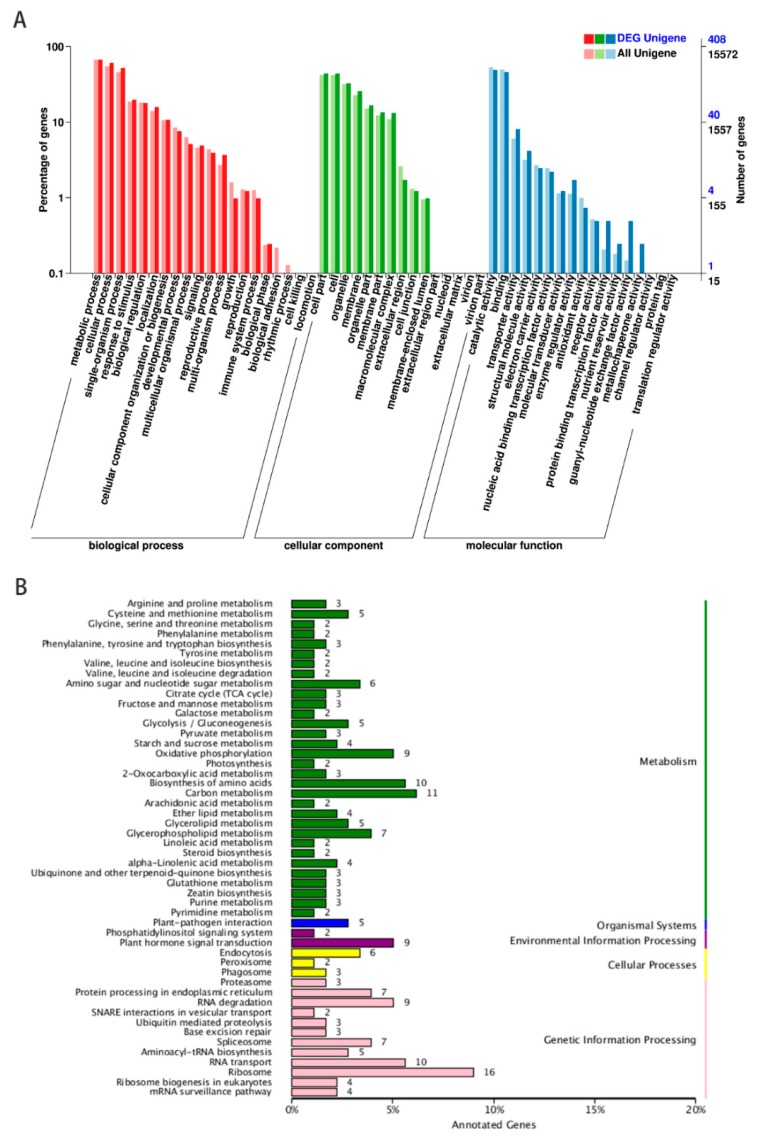
GO terms and KEGG pathway enrichment in tomato RKNT and Sneb821T. (**A**) Gene categories disctribution of lncRNAs under molecular functions (481 genes), cellular components (790 genes), and biological processes (1115 genes). (**B**) KEGG pathways enrichment of neighboring genes of lncRNAs in cellular processes (11 genes), environmental information processing (11 genes), genetic information processing (73 genes), metabolism (120 genes) and organismal systems (5 genes).

**Figure 4 ijms-21-00911-f004:**
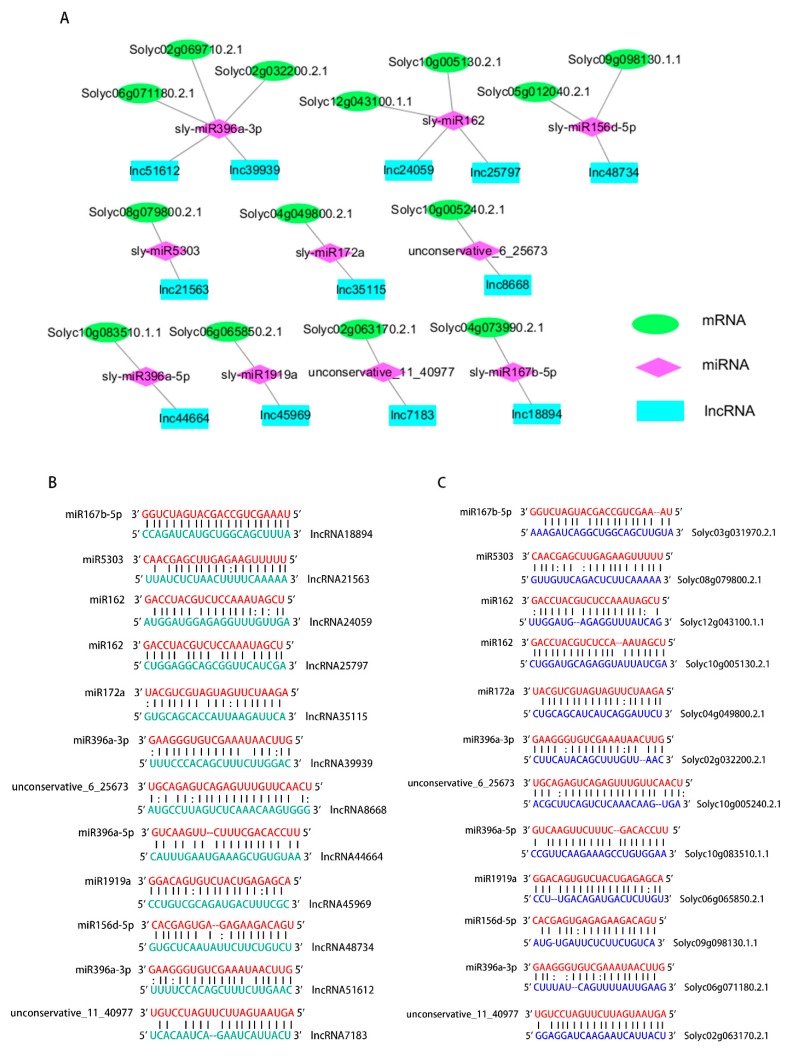
Predicted miRNA targets and endogenous target mimcs from lncRNAs and mRNAs. (**A**) ceRNA network of the lncRNAs, miRNAs and mRNAs in tomato. The green sequences are tomato mRNAs.The red sequences are tomato miRNAs. The blue sequences are tomato lncRNAs. (**B**) The lncRNAs act as miRNA targets. (**C**) The mRNAs act as endogenous target mimics of miRNAs. The green sequences are tomato lncRNAs. The red sequences are tomato miRNAs. The blue sequences are tomato mRNAs. All mRNAs, miRNAs and lncRNAs were assessed using psRNATarget and psRobot.

**Figure 5 ijms-21-00911-f005:**
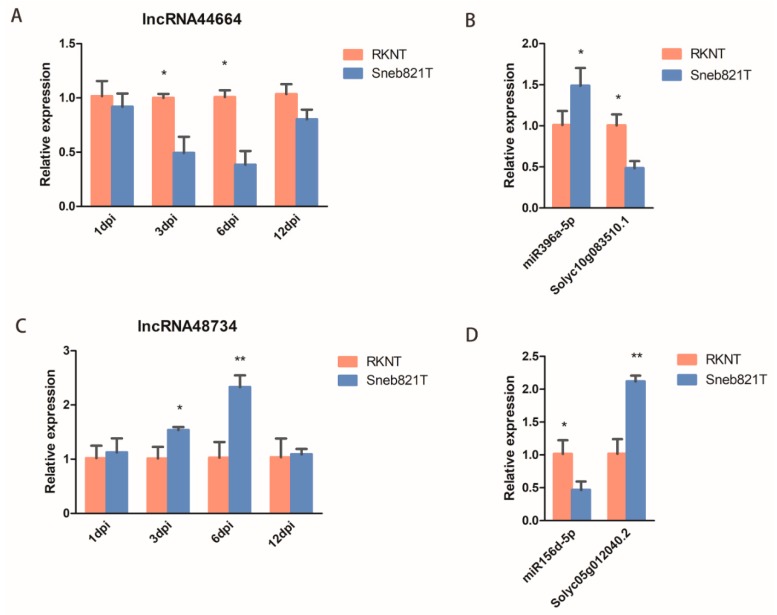
Verification of differentially expressed lncRNAs by qRT-PCR. (**A**) qRT-PCR analysis of lncRNA44664 in tomato plants. (**B**) qRT-PCR analysis of miR396a and the target genes in tomato plants. (**C**) qRT-PCR analysis of lncRNA48734 in tomato plants. (**D**) qRT-PCR analysis of miR156d and the target genes in tomato plants. All data are the means ± SD of three independent experiments. * indicate a significant difference at the *p* = 0.05 level, ** indicate a significant difference at the *p* = 0.01 level.

**Figure 6 ijms-21-00911-f006:**
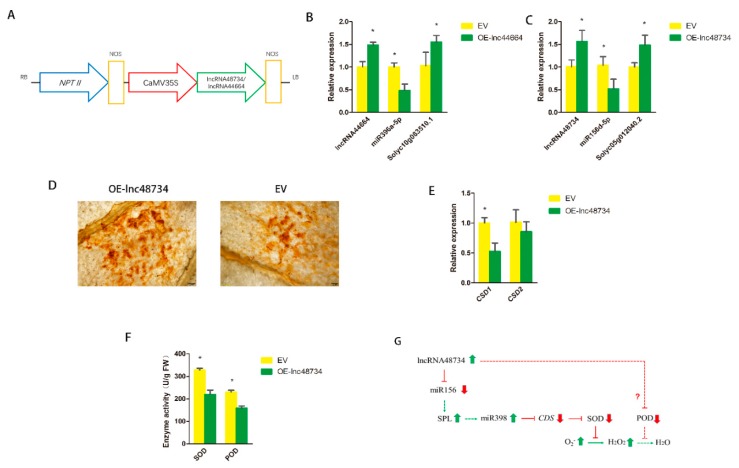
Function verification of lncRNAs. (**A**) Schematic diagram of the gene cassette containing lncRNA48734 and lncRNA44664. (**B**) qRT-PCR analysis of lncRNA44664, miR396a and the target genes in the EV (empty vector) and OE-lnc44664 tomato plants. (**C**) qRT-PCR analysis of lncRNA48734, miR156d and the target genes in the EV and OE-lnc48734 tomato plants. (**D**) DAB staining of the leaves of EV and OE-lnc48734 tomato plants for H_2_O_2_ after inoculation with *Agrobacterium* after 3 days. Scale bars = 200 μm. (**E**) qRT-PCR analysis of *CSD1* and *CSD2* in tomato plants. (**F**) The enzyme activity of SOD and POD of EV and OE-lnc48734 tomato plants. (**G**) The predictive model of lncRNA48734-ROS accumulation interaction network involved in tomato. Arrows indicate positive regulation, and blunt-ended bars indicate inhibition. All data are the means ± SD of three independent experiments. * indicate a significant difference at the *p* = 0.05 level.

**Figure 7 ijms-21-00911-f007:**
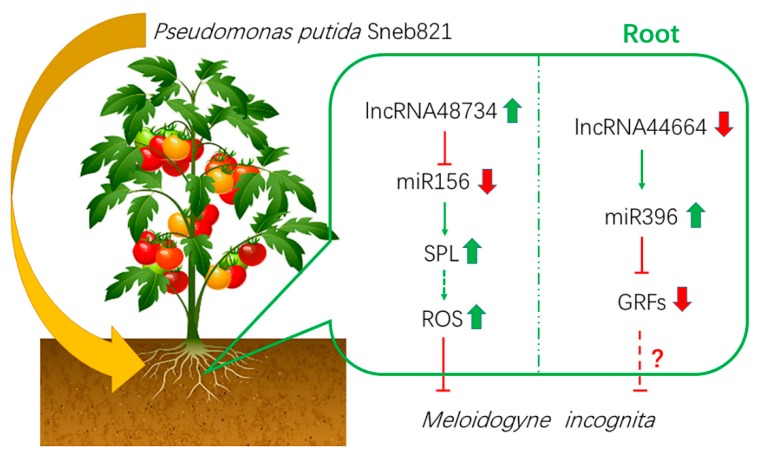
Model of biocontrol bacteria induces tomato resistance to *M. incognita*. Arrows indicate positive regulation, and blunt-ended bars indicate inhibition. A line does not necessarily represent unique or direct regulation. A question mark refers to unverified regulation under *M. incognita* stress.

**Table 1 ijms-21-00911-t001:** Raw reads were produced from six cDNA libraries of tomato samples.

Sample	Clean Reads	Mapped Reads	Mapped Ratio
RKNT 1	123068008	108742693	88.36%
RKNT 2	118020220	105048103	89.01%
RKNT 3	129145372	114727340	88.84%
Sneb821T1	115966444	104203875	89.86%
Sneb821T 2	126689802	114918861	90.71%
Sneb821T 3	109378368	99030869	90.54%

**Table 2 ijms-21-00911-t002:** GO terms and KEGG pathways for the target genes of selected 12 lncRNAs.

lncRNA	mRNA	GO Terms	KEGG Pathway
lncRNA18894	Solyc04g073990.2.1	transferase activity	-
lncRNA21563	Solyc08g079800.2.1	-	plant hormone signal transduction
lncRNA24059	Solyc12g043100.1.1	-	endocytosis
lncRNA25797	Solyc10g005130.2.1	DNA binding	-
lncRNA35115	Solyc04g049800.2.1	transcription cofactor activity	-
lncRNA39939	Solyc02g069710.2.1	-	ribosome
	Solyc02g032200.2.1	-	plant-pathogen interaction
lncRNA8668	Solyc10g005240.2.1	-	biosynthesis of amino acids
lncRNA44664	Solyc10g083510.1.1	cell proliferation	-
lncRNA45969	Solyc06g065850.2.1	-	biosynthesis of amino acids
lncRNA48734	Solyc05g012040.2.1	-	oxidative phosphorylation
	Solyc09g098130.1.1	-	plant-pathogen interaction
lncRNA51612	Solyc06g071180.2.1	-	spliceosome
lncRNA7183	Solyc02g063170.2.1	cell death	-

**Table 3 ijms-21-00911-t003:** Annotation of 12 selected target genes of lncRNAs, miRNAs and mRNA.

lncRNA	miRNA	mRNA	Annotation
lncRNA18894	miR167b-5p	Solyc04g073990.2.1	Annexin
lncRNA21563	miR5303	Solyc08g079800.2.1	Growth-regulating factor 12
lncRNA24059	miR162	Solyc12g043100.1.1	Pentatricopeptide repeat-containing protein
lncRNA25797	miR162	Solyc10g005130.2.1	Dicer double-stranded RNA-binding fold
lncRNA35115	miR172a	Solyc04g049800.2.1	AP2-like ethylene-responsive transcription factor
lncRNA39939	miR396a-3p	Solyc02g069710.2.1	Sodium/calcium exchanger protein
		Solyc02g032200.2.1	Tir-lrr, resistance protein fragment
lncRNA8668	unconservative_6_25673	Solyc10g005240.2.1	Transcription factor Myb
lncRNA44664	miR396a-5p	Solyc10g083510.1.1	Growth-regulating factor 2
lncRNA45969	miR1919a	Solyc06g065850.2.1	Flotillin domain protein
lncRNA48734	miR156d-5p	Solyc05g012040.2.1	Squamosa promoter binding protein 3
		Solyc09g098130.1.1	Cc-nbs-lrr, resistance protein
lncRNA51612 lncRNA7183	miR396a-3p unconservative_11_40977	Solyc06g071180.2.1 Solyc02g063170.2.1	Dynein light chain 1 cytoplasmic Homology

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
