# Peer review of "Functional Analysis of Long Non-Coding RNAs Reveal Their Novel Roles in Biocontrol of Bacteria-Induced Tomato Resistance to Meloidogyne incognita"

_ijms, 2020, doi:10.3390/ijms21030911_

Round 1
Reviewer 1 Report
The paragraphs need to be modified in many places throughout the manuscript. I have mentioned a few lines below.
Title: …….biocontrol of bacteria-induced…..
Line 59: ……sites, which…..
Line 86: The results showed that the disease resistance mechanism of Sneb821 induced resistance.???????
Line 91 and elsewhere: “M. incognita” needs to be italicized.
Lines 112-115: confusing.
Author Response
Response to Reviewer 1 Comments
Point 1: Title: …….biocontrol of bacteria-induced…..
Response 1: Title: Functional analysis of long non-coding RNAs reveal their novel roles in biocontrol of bacteria-induced tomato resistance to M. incognita
Point 2: Line 59: ……sites, which…..
Response 2: lncRNAs act as a miRNA sponge that contain miRNA-binding sites, which have been indicated to regulate miRNAs and their targets.
Point 3: Line 86: The results showed that the disease resistance mechanism of Sneb821 induced resistance.???????
Response 3: The results showed that the disease resistance mechanism of Sneb821 is induced resistance.
Point 4: Line 91 and elsewhere: “M. incognita” needs to be italicized.
Response 4: Two experiments were set up to explore the inducement mechanism of Sneb821 in two treated groups including inoculation of M. incognita treated plants (RKNs treatment, RKNT) group and inoculation of M. incognita treated plants after Sneb821 irrigation 5 days (Sneb821 treatment, Sneb821T) group.
Point 5: Lines 112-115: confusing.
Response 5: The mapped reads and mapped ratio of RKNT1, RKNT2, RKNT3 were (108742693 (88.36 %), 105048103 (89.01 %), 114727340 (88.84 %), respectively). Sneb821T1, Sneb821T2, Sneb821T3 were mapped (104203875 (89.86% ), 114918861 (90.71 %), 99030869 (90.54 %) respectively) and reads to the reference genome of tomato.
Reviewer 2 Report
Thanks for the interesting read. In my opinion the MS is well presented and conducted as well as the discussion and conclusion sections that are well supported by the data presented.
Great work!
Minor concerns:
Please only revise the scientific names that should be reported in Italic along the MS.
line 353: Are five days sufficient for the efficient root infection and colonization by M. incognita? The authors quantify the infection rate in the inoculated samples by means of PCR/qPCR or by microscopic techniques?
Author Response
Response to Reviewer 2 Comments
Point 1: Please only revise the scientific names that should be reported in Italic along the MS.
Response 1: All the scientific names have been changed in Italic, such as M. incognita.
Point 2: line 353: Are five days sufficient for the efficient root infection and colonization by M. incognita? The authors quantify the infection rate in the inoculated samples by means of PCR/qPCR or by microscopic techniques?
Response 2: Five days are sufficient for root infection and it is at the beginning of the infection. We quantify the infection rate by microscopic examination.
Reviewer 3 Report
The article demonstrated that the treatment of Pseudomonas putida Sneb821 strain could regulate expression of long non-conding RNA (lncRNA) tightly associated with resistance response of tomato plants against Meloidogyne incognita infection. Through a comprehensive analysis of lncRNA in infected tomato plants with the nematode, the authors showed two different regulatory networks for induced resistance by the strain against root-not nematode infection. Experimental designs and data analysis were very sound. Thus I believe that the article will give the audience a new insight into the mechanism of induced resistance in plants. However, the current version of the manuscript should have to be revised to make their argument more clear.
Title: Functional analysis of long non-coding RNAs reveals ...... An analysis is a singular noun. The scientific names have to be italicized in the manuscript. Please double-check the errors in the overall text. In Figure 2C: Is a title of Y-axis H2O2, not O2-, production, isn't it? Line 114. they should have to cite the version of the reference genome of tomato in the text. Line 156-159. The caption of figure 3 can explain the figure in detail. For example, what is the number of bars in Figure 3B? Even if they explain the meaning of the numbers in the text, I think that the authors should have to describe it in the caption. Moreover, the title of figure 3 has to be rephrased. Line 193-197 and line212-216. I ask the authors to add a few sentences related to the analytical methods in the captions. Line 366. Four, not three. ? My main criticism is that detailed explanation or hypothesis about the way regulation of lncRNA expression by the P. putida Sneb821 treatment. I also think that the mechanism is still an open question. However, they have to try to describe the mechanism. The second one is that logic to select 12 lncRNA is not enough.
